# An evaluation of trace elements and oxidative stress in patients with benign paroxysmal positional vertigo

Hüseyin Günizi[1]*, Hasan Basri Savaş[2]

1 Department of Otolaryngology, Alkü Alanya Alaaddin Keykubat University Medical School, Alanya, Antalya, Turkey, 2 Department of Clinical Biochemistry, Alkü Alanya Alaaddin Keykubat University Medical School, Alanya, Antalya, Turkey

☯ These authors contributed equally to this work.
* drgunizi@gmail.com

**Data Availability Statement:** All relevant data are within the paper and its Supporting Information files.

## Abstract

### Objectives

Vertigo and Dizziness are a common complaint among the reasons for applying to the ear nose throat clinic. Benign Paroxysmal Positional Vertigo (BPPV) is the most common cause of perpheric vertigo. Oxidative stress is caused by the formation of hydroxyl radicals, superoxide anions and hydrogen peroxide, which are reactive oxygen derivatives (ROS). The aim of this study is to investigate the relationship between complaints and serum trace element and oxidative stress levels in patients with BPPV.

### Methods

This study was conducted with 66 adult patients who presented to the ENT policlinic with the complaint of vertigo and were diagnosed with BPPV between May 2020 and September 2020. Blood samples were taken from patients diagnosed with BPPV to measure serum Zn and Cu levels and oxidative stress levels during an attack.

### Results

The mean ages of the study patients and healthy controls were 45.7 ± 15.1 and 44.7±13.2. Female / Male ratio were 28(42.5%)/38(57.5%) and 32(48.5%)/34(51.5%) in study and control group. We found serum Cu levels were lower in the patient group (p <0.05). Serum Total Thiol and Native Thiol values were lower in patients with BPPV. Total Thiol results were statistically significant.(p<0.05) Disulfide values were significantly higher in the disease group. (P <0.05). Thiol Oxidized / Thiol Reduced ratio (2243.6±6.7/343.8±125.3) was higher in control group. (p<0.05)

### Conclusion

Serum oxidative stress and trace elements play a role in the pathophysiology of BPPV. We present the cut-off values for Cu and Zn in vertigo patients for the first time in the literature.

**Funding:** The funders had no role in study design, data collection and analysis, decision to publish, or preparation of the manuscript.

**Competing interests:** The authors have declared that no competing interests exist.

We think that these cut-off values of the trace elements and thiol/disulfide hemostasis can be used clinically by physicians in the etiology, diagnosis and treatment of vertigo.

## Introduction

Vertigo and Dizziness are a common complaint among the reasons for applying to the ear nose throat clinic [1]. It affects approximately 20–30% of the general population [2]. Vertigo is an acute symptom arising from the asymmetry of the vestibular system in single or recurrent attacks. Vertigo may be due to peripheral or central diseases. Benign Paroxysmal Positional Vertigo (BPPV) is the most common cause of perpheric vertigo [3]. Among the peripheral vestibular disease, the frequency of BPPV has been reported to be 25–40%. BPPV is ascribed to otoconial matter dislodged from utricular macula and attached to the cupula of the affected semicircular canal (cupulolithiasis) or free-floating within its lumen (canalolithiasis) [4,5]. In BPPV, dizziness develops due to otoliths in the posterior (85–90%) and horizontal (5–15%) canals [3].

Oxidative stress is a condition caused by an imbalance between production and accumulation of oxygen reactive species (ROS) in cells or tissues and the ability of a biological system to detoxify these reactive products. Superoxide radicals, hydrogen peroxide, hydroxyl radicals and singlet oxygen are commonly defined reactive oxygen species (ROS) [6,7]. In the tissues, high rates of oxygen utilization lead to the generation of partially reduced forms of oxygen or reactive oxygen species (ROS), which are derived from the activity of electron transport chain in mitochondria [8]. Free radicals cause oxidative cell damage and this is a situation that occurs continuously during metabolic processes. ROS and free radicals react with and damage several cellular structures such as proteins, lipids, membranes, lipoproteins, and deoxyribonucleic acid [9]. ROS react with most biomolecules, their major targets include unsaturated lipids and intracellular thiols [8]. In oxidative stress, thiol oxidizes and converts to disulfide. Thiol balance is an important and dual indicator of oxidant-antioxidant balance [10]. Prevention of oxidative damage, keeping it in balance and maintaining vitality is only possible with antioxidant activity. If this process gets out of control it can induces several chronic inflammatory diseases as rheumatoid arthritis, and degenerative, body aging process and acute pathologies as stroke [9,11].

Trace elements constitute the building blocks of enzymes, hormones and vitamins as well as metabolism support, healthy tissue formation and supporting the immune system. Zn and Cu are the trace elements that are essential and vital for the body. Zinc is an enzymatic activator and catalyst in hormonal processes and amino acid cycle. Copper is a necessary trace element in the controlled work of enzymes in biological oxidation events. The extracellular form of the superoxide dismutase enzyme, which is the only antioxidant secreted by fibroblast, glia and endothelial cells and capable of inactivating superoxide radicals enzymatically at the extracellular level, contains Cu and Zn. It has a very important role in the prevention of many diseases such as oxidant damage, inflammation and fibrosis [12]. Therefore, the antioxidant effect of Zn and Cu is remarkable.

Disruption of the balance between oxidative stress and antioxidant system plays a role in the etiopathogenesis of many diseases. There are not enough studies investigating the link between oxidative stress level, serum Zn/Cu levels and symptoms in vertigo patients. The aim of this study is to investigate the relationship between complaints and serum trace element and oxidative stress levels in patients with a diagnosis of BPPV.

## Materials and methods

This study was conducted with 66 adult patients who presented to the ENT policlinic with the complaint of vertigo and were diagnosed with BPPV between May 2020 and September 2020 in Alanya Alaaddin Keykubat University Hospital. Patients with a history of Meniere's disease, vestibular neuritis, acoustic neurinoma, sudden hearing loss, hearing loss on audiometric examination, history of head trauma, and having an otologic surgery were not included in the study. Detailed ear, nose and throat and neurological examinations of all patients were performed. An audiometric examination was performed, patients with hearing loss were excluded from the study. Dix-Hallpike and Supine Head-Roll maneuvers were performed. Epley's and Barbecue repositioning maneuver was applied to patients who developed rotatory nystagmus during the maneuvers and supported BPPV. Among these patients, those who did not have diabetes mellitus, hyperlipidemia, etc., had normal parameters such as sedimentation and c-reactive protein in routine blood tests, and had a first BPPV attack were included in the study. Blood samples were taken from patients diagnosed with BPPV to measure serum Zn and Cu levels and oxidative stress levels during at the first attack and before 12:00 a.m. while they were fasting. The control group included 66 healthy volunteers with age and gender distribution similar to the BPPV patient group. They had no neurotologic symptoms and no history of any dizziness or imbalance.

This study was approved by the Local Ethical Committee of Alanya Alaaddin Keykubat University (Protocol Number: 18–4: Approve Date:22.04.2020). Written consent was obtained from all patients in our study.

### Measurement of thiol disulfide homeostasis

Thiol / disulfide homeostasis measurements were made using the standard colorimetric method described in the literature [10]. Two separate serum samples were collected for each patient. Serum total thiol measurement was performed using standard colorimetric method and commercial kit (Rel Assay Diagnostics, Gaziantep). Ten μL sample was treated with 10μL sodium borohydride in 50% methanol-water solution (v/v; R1). 110μL 6.715mM formaldehyde and 10.0mM ethylenediaminetetraacetic acid (EDTA) in Tris buffer 100mM (pH 8.2) used for Excess reductants eliminating to determine total thiol. A 10μL sample was treated with 10μL sodium chloride in 50% methanol-water solution (v/v; R1') and 110μL 6.715mM formaldehyde and 10.0mM EDTA in Tris buffer 100mM (pH 8.2) used to determine native thiol. Dynamic disulfide amount was calculated by taking half of the difference between the total thiol and natural thiol groups. All the chemicals were purchased from Merck Chemicals (Darmstadt, Germany) and Sigma-Aldrich Chemie (Milwaukee, Wisconsin, USA). In this way, natural and total thiols were calculated. Then serum disulfide levels and disulfide / natural thiol, disulfide / total thiol and natural / total thiol ratios were determined. Units are given as μmol/L [10].

### Zinc-copper levels

Serum zinc–copper levels were measured using a standard colorimetric method and a commercial kit (Rel Assay Diagnostics, Gaziantep). Units are given as μg / dl [13,14].

### Statistical analysis

The distribution of the data in the study was evaluated with ShapiroWilk's test statistics, histogram, and q-q graph.The independent two sample t test and Mann-Whitney U test were used to compare the difference between the means of two independent groups. Bonferroni correction was not applied because we only looked at 2 parameters. Categorical variables were

expressed as numbers and percentages, whereas continuous variables were summarized as mean and standard deviation or median (minumum-maximum). Roc analysis was applied for all parameters. The cut-off values, area under the curve (AUC), sensitivity and specificity were calculated using the receiver operating characteristic (ROC) curve technique for trace elements and oxidation products. Chi-square analysis was performed to determine the relationships between categorical variables. Pearson correlation analysis was performed to determine the direction and strength of the relationships between trace elements and oxidation products. Univariate binary logistics regression analysis was performed to determine the risk factors affecting BPPV. Data analysis was performed in TURCOSA Cloud (Turcosa Ltd Co, www.turcosa.com.tr) statistics software. It was accepted as the statistical significance level ($p < 0.05$).

## Results

In our study, 66 patients diagnosed with BPPV and 66 healthy control groups were examined prospectively. The mean ages of the study patients and healthy controls were 45.8 ± 15.1 and 44.8±13.2. Female / Male ratio were 28(42.5%)/38(57.5%) and 32(48.5%)/34(51.5%) in study and control group. There were 54 (82%) patients with BPPV affecting the posterior and 12 (18%) patients with BPPV affecting the lateral canal. Serum Cu and Zn levels of patients with BPPV were compared with the control group. There were no statistically significant differences between male and female genders in terms of Cu and Zn levels. We found mean serum Cu levels to 119,6± 47.6 μg/dl in the patient group, and this result was lower than control group. Low Cu level was statistically significant (p <0.05) as indicated in Table 1. Serum mean zinc level was higher in study patients than control patients (87.1± 13.2 μg/dl). Serum Total Thiol (611.7±84.8/535.0±154.7)and Native Thiol (551.2±89.3/328.4±98.1) values were higher in the control group. These results were statistically significant. (P <0.05) We found high disulfide values in BPPV patients, and these results were statistically significant.(p <0.05) Reduced Thiol (89.9±4.9) level was higher in control group and Oxidized Thiol(19.3±4.1) level was higher in BPPV group.(p<0.05) Thiol Oxidized / Thiol Reduced ratio (2243.6±6.7/343.8 ±125.3) was higher in control group. (p<0.05) as indicated in Table 2.

ROC analysis of trace elements and oxidation products, Univariate Logistic regresson analysis, Pearson correlation matrix of trace elements and oxidation products for control group, Pearson correlation analysis of trace elements and oxidation products for control BPPV were given below as indicated in Table 3 and Fig 1, Tables 4–6. In the ROC analysis of Cu, Zn and Total Thiol values, the AUC values were below 0.75, so we could not evaluate these values as independent predictive factors. According to Univariate Logistic regresson analysis; Those with a serum Zn level below 81.7 μg/dL have a 2,210 fold higher risk of developing vertigo than those with an age parameter above 81.7 μg/dL.(Odds ratio: 2.210; %95 CI: 1.035–4.718, p = 0.040). Those with a serum Cu level below 104.6 μg/dL have a 2,210 fold higher risk of developing vertigo than those with an age parameter above 104.6 μg/dL.(Odds ratio: 2.968, % 95 CI: 1.445–6.100 p = 0.003). The risk of developing BPPV in patients with serum NT level

**Table 1. Comparison of trace elements between study groups.**

| Variables | BPPV | Control | P |
|---|---|---|---|
| Age | 45.8±15.1 | 44.8±13.2 | 0.692 |
| Zn(μg/dL) | 87.1±13.2 | 85.8 ±7.6 | 0.496 |
| Cu(μg/L) | 119 ± 47.6 | 135.7± 42.2 | 0.041 |

Zn; zinc, Cu; copper, BPPV; Benign Paroxysmal Positional Vertigo.

**Table 2. Comparison of thiol, native thiol and disulphide.**

|  | BPPV | Control | P |
|---|---|---|---|
| **Total Thiol(μmol/l)** | 535.0±154.7 | 611.7±84.8 | <0.001 |
| **Native Thiol(μmol/l)** | 328.5±98.1 | 551.2±89.3 | <0.001 |
| **Disulphide (μmol/l)** | 104.2±41.2 | 30.2±13.7 | <0.001 |
| **Reduced Thiol(μmol/l)** | 61.7±9.5 | 89.9±4.9 | <0.001 |
| **Oxidized Thiol(μmol/l)** | 19.3±4.1 | 5.0±2.4 | <0.001 |
| **Thiol Oxidation/Reduction Ratio** | 419.7(105.5–4459.6) | 1359.6(616.1–4459.6) | <0.001 |

Data were expressed as arithmatic mean and standard deviation or median (minumum-maximum). BPPV; Benign Paroxysmal Positional Vertigo.

lower than 416.5 μmol/L was 85,615 times higher than those with serum NT level higher than 416.5 μmol/L (Odds ratio: 85.615% 95 CI:23.160–316.499 P<0.001).

According to Roc analysis, the results of AUC values were Oxidized Thiol, Thiol Oxidized / Thiol Reduced ratio, Reduced Thiol, Disulfid and Native Thiol, Total Thiol, CU and Zn, in order of statistical significance. Cut-off and p values according to this order 554.4 μmol/l-p<0.001, 13.3 μmol/lp<0.001, 73.5 μmol/lp<0.001, 60.4 μmol/lp<0.001, 416.5 μmol/lp<0.001, 552.4 μmol/lp It was <0.001,104.6 μg/Lp<0.001, 81.7 μg/Lp<0.001. When the areas under the curve (AUC) were evaluated in Roc analysis, Oxidizel Thiol, Thiol Oxidized / Thiol Reduced ratio, Reduced Thiol, Disulfid and Native Thiol were found to be significantly higher.

## Discussion

Researchers previously stated that oxidative stress has an effect on the etiology of hearing loss, rhinosinusitis, otitis media, chronic tonsillitis and laryngeal cancer [15,16]. There are studies explaining that the inflammatory process and oxidative stress formation play a role in the

**Table 3. ROC analysis of trace elements and oxidation products.**

| Variables | AUC-ROC(%95 CI) | P value | Cut-off value | Sensitivity | Specificity | +LR | -LR | PP | NP |
|---|---|---|---|---|---|---|---|---|---|
| **Zn (μg/dL)** | 0.508(0.419–0.596) | <0.001 | 81.7 | 0.424 | 0.757 | 1.75 | 0.76 | 0.64 | 0.57 |
| **Cu (μg/L)** | 0.648(0.560–0.729) | <0.001 | 104.6 | 0.591 | 0.712 | 2.05 | 0.57 | 0.67 | 0.64 |
| **Total Thiol(μmol/l)** | 0.737(0.652–0.809) | <0.001 | 552.4 | 0.712 | 0.803 | 3.62 | 0.36 | 0.78 | 0.74 |
| **Native Thiol(μmol/l)** | 0.940(0.885–0.974) | <0.001 | 416.5 | 0.818 | 0.954 | 18.0 | 0.19 | 0.95 | 0.84 |
| **Disulphide(μmol/l)** | 0.996(0.965–0.999) | <0.001 | 60.4 | 0.969 | 0.984 | 64.0 | 0.03 | 0.99 | 0.97 |
| **Reduced Thiol(μmol/l)** | 0.984(0.945–0.998) | <0.001 | 73.5 | 0.970 | 1.000 | - | 0.03 | 1.000 | 0.97 |
| **Oxidized Thiol(μmol/l)** | 0.999(0.971–1.000) | <0.001 | 13.3 | 0.984 | 1.000 | - | 0.02 | 1.00 | 0.99 |
| **Thiol Oxidation / Reduction Ratio** | 0.999(0.970–1.000) | <0.001 | 554.4 | 0.970 | 1.000 | - | 0.03 | 1.00 | 0.97 |

AUC-ROC; area under ROC curve, PP; positive predictive value, NP; negative predictive value, 95% CI; confidence interval, +LR; positive likelihood ratio, -LR; negative likelihood ratio.

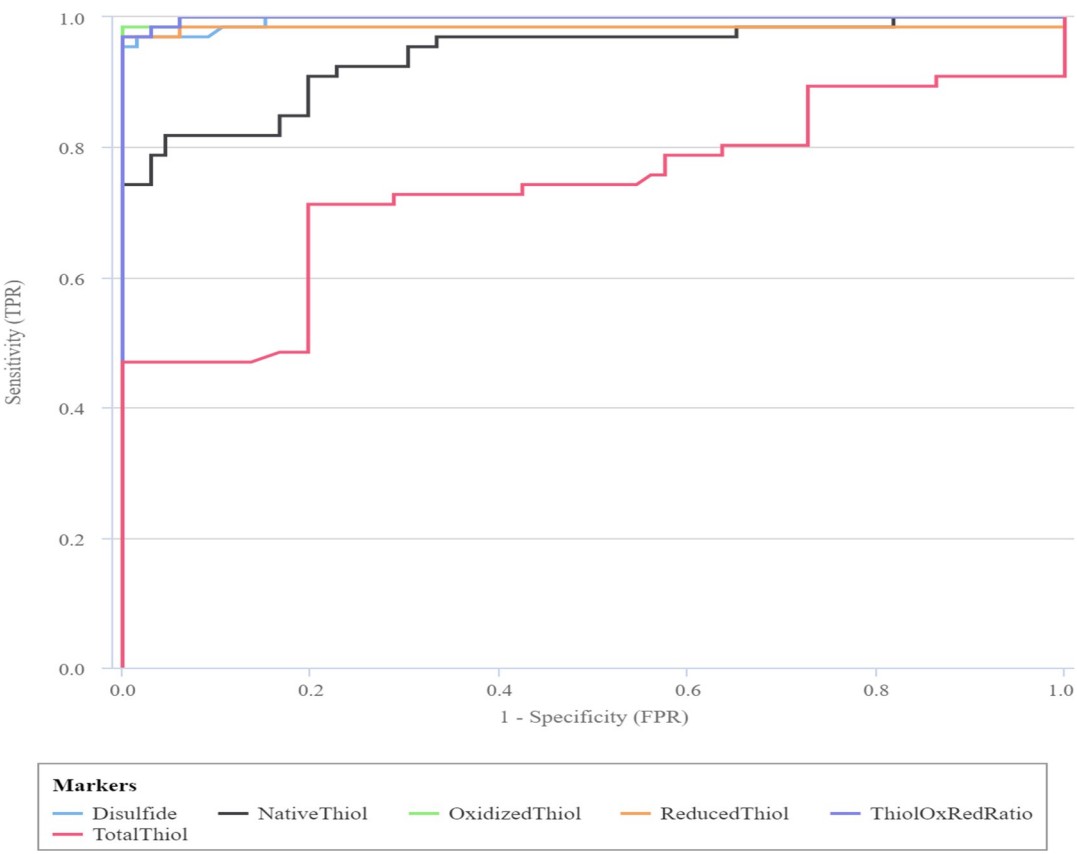

**Fig 1. ROC curve of Disulfide, Native Thiol, Oxidized Thiol, Reduced Thiol and Total Thiol.**

development of vertigo [17]. İn a study, researchers indicated that calcium metabolism and its relationship with oxidative stress may play a role in the development of BPPV [18].

Oxidative stress is effective in calcium metabolism. The endoplasmic reticulum is the major organelle in calcium storage and has the ability to increase calcium flux under stress. This situation increases the formation of ROS in mitochondria. Calcium influx to the mitochondria and increased oxidative stress cause rupture of the outer mitochondrial membrane and apoptosis. Physiopathology of BPPV occurs as a result of endolymph flow due to free otoliths escaping from the utricular macula to the semicircular canal [19]. Calcium carbonate constitutes the structure of otoliths. Increasing the amount of free calcium in the endolymph decreases the dissolution of the otolith escaping into the semicircular canal in the endolymph. This increased calcium level by oxidative stress prevents the reabsorption of otoliths escaping to the semicircular canal. Zucca et al. In their study with the frog inner ear, they reported that in order to dissolve the otoconia in the endolymph, the calcium level in the endolymph should be low, and on the contrary, if the calcium in the endolymph increases, the otoconi cannot be resolved. They stated that high calcium provides a long duration of BPPV attack [20]. This fact may contribute to explain the role of increased calcium in the inner ear due to oxidative stress in the pathophysiology of BPPV. In a study, they found that serum vitamin D levels in BPPV patients were significantly lower than in the control group. They suggested that oxidative stress increased and BPPV developed in patients with vitamin D deficiency due to the relationship of vitamin D to Ca metabolism [21]. The fact that vitamin D is directly related to calcium physiology makes our hypothesis strong. However, in our study, we could not measure serum calcium

**Table 4. Univariate Logistic regresson analsis.**

| Variables | Groups | | p | Univariate Binary Logistic Regression Analysis | | p |
|---|---|---|---|---|---|---|
| | Control n(%) | BPPV n(%) | | OR | 95%CI | |
| Zn 81.7≥ | 51(77.3) | 40(60.6) | 0.039 | Reference | | 0.040 |
| 81.7< | 15(22.7) | 26(39.4) | | 2.210 | 1.035–4.718 | |
| Cu 104.6≥ | 47(71.2) | 30(45.5) | 0.003 | Reference | | |
| 104.6< | 19(28.8) | 36(54.5) | | 2.968 | 1.445–6.100 | 0.003 |
| Total thiol 552.4≥ | 53(80.3) | 20(30.3) | <0.001 | Reference | | |
| 552.4< | 13(19.7) | 46(69.7) | | 9.377 | 4.204–20.915 | <0.001 |
| Native Thiol 416.5≥ | 63(95.5) | 13(19.7) | <0.001 | Reference | | |
| 416.5< | 3(4.5) | 53(80.3) | | 85.615 | 23.160–316.499 | <0.001 |
| Disulphide 60.4≥ | 1(1.5) | 64(97.0) | <0.001 | | | |
| 60.4< | 65(98.5) | 2(3.0) | | - | | |
| Reduced Thiol 73.5≥ | 66(100.0) | 3(4.5) | | | | |
| 73.5< | 0(0.0) | 63(95.5) | <0.001 | - | | |
| Oxidized Thiol 13.3≥ | 0(0.0) | 65(98.5) | <0.001 | | | |
| 13.3< | 66(100.0) | 1(1.5) | | - | | |
| Thiol Oxid./Red. 554.4≥ | 66(100.0) | 3(4.5) | <0.001 | | | |
| 554.4< | 0(0.0) | 63(95.5) | | - | | |

Univariate Logistic regresson analsis: -; OR could not be calculated because frequencies were insufficient, 95% CI; confidence interval, OR; Odss Ratio Thiol Oxid./Red.; Thiol oxidation / reduction ratio.

and vitamin D values in our patients due to financial reasons. We think that the increased amount of calcium in the endolymph due to oxidative stress is important in the development of BPPV.

The significant relationship between the concentration of serum oxidative stress markers such as SOD, Native Tyhol, Disulfide and BPPV has been explained [15,17,22,23]. In oxidative stress, thiol oxidized and turns into disulfide. Thiol balance is an important and dual indicator of oxidant-antioxidant balance [10]. Thiol / disulfide homeostasis (TDH) has an important

**Table 6. Pearson correlation matrix of trace elements and oxidation products for BPPV group.**

| | Zn | Cu |
|---|---|---|
| **Total Thiol** | r = 0.5899 p = <0.001 | r = 0.8719 p = <0.001 |
| **Native Thiol** | r = 0.5916 p = <0.001 | r = 0.8734 p = <0.001 |
| **Disulfide** | r = 0.3857 p = 0.001 | r = 0.5869 p = <0.001 |
| **Reduced Thiol** | r = 0.0725 p = 0.563 | r = 0.1001 p = 0.424 |
| **Oxidized Thiol** | r = -0.1197 p = 0.338 | r = -0.1351 p = 0.279 |
| **Thiol Oxidation / Reduction Ratio** | r = 0.0958 p = 0.444 | r = 0.0946 p = 0.450 |

The top side of the matrix is the pearson correlation coefficient (r) and the bottom side is the significance value (p). BPPV; Benign Paroxysmal Positional Vertigo.

**Table 5. Pearson correlation matrix of trace elements and oxidation products for control group.**

| Variables | Zn | Cu |
|---|---|---|
| **Total Thiol** | r = 0.2158<br>p = 0.082 | r = 0.9049<br>p = <0.001 |
| **Native Thiol** | r = 0.2068<br>p = 0.096 | r = 0.8367<br>p = <0.001 |
| **Disulfide** | r = -0.0065<br>p = 0.959 | r = 0.073<br>p = 0.560 |
| **Reduced Thiol** | r = 0.0934<br>p = 0.456 | r = 0.2015<br>p = 0.105 |
| **Oxidized Thiol** | r = -0.0871<br>p = 0.487 | r = -0.1961<br>p = 0.115 |
| **Thiol Oxidation / Reduction Ratio** | r = 0.0081<br>p = 0.949 | r = 0.047<br>p = 0.708 |

The top side of the matrix is the pearson correlation coefficient (r) and the bottom side is the significance value (p).

role in antioxidant processes, cell growth and apoptosis [10,24]. It is thought that the imbalance in TDH plays a role in the pathogenesis of diseases through oxidative stress and tissue inflammation. In the literature, the effect of TDH in many diseases such as sudden sensorineural hearing loss, vitiligo, Hashimoto thyroiditis, and polycystic ovary syndrome has been investigated [25,26]. In studies with patients with coronary atherosclerosis, it was found that the severity of atherosclerosis increased in patients with low serum NT/disulfide ratio [27]. Low NT levels were found to be significantly associated with the disease in the patient group diagnosed with nasal polyposis [28]. However, there are very few studies on BPPV and TDH in the literature. Şahin E. et al. found significantly lower SH and high SD rates during an attack. They also found lower NT levels, lower NT / TT ratio and higher disulfide levels in patients with BPPV in the control blood [23]. Li et al. found high SOD level after the vertigo attack [18]. With the support of these findings, the relationship between oxidative stress and BPPV attack is significant. In our study, we found significantly lower NT level, lower NT / TT ratio and higher disulfide level in the BPPV group compared to the control group during the attack (p <0.05). Although native thiol elevation appears to be a risk factor for vertigo in ROC analysis native thiol is a part of the thiol-disulphide balance. For this reason, it would be more appropriate to evaluate the balance in its entirety. When we look at the comparison between the groups, it is seen that the disulfide levels, which can be an indicator of the relevant balance, are lower in the control group. As a result, when the thiol-disulphide balance is considered, high disulphide levels appear as a risk factor for vertigo. These findings were consistent with the literature.

Antioxidants prevent the formation of reactive oxygen species and prevent the damage caused by these substances and provide detoxification. Antioxidants prevent the progression of autooxidation by reacting rapidly with radicals [10]. Zn and Cu, which are cofactors in many metalloenzymes, are also very powerful antioxidants and they are essential elements for antioxidation. Superoxide dismutase (SOD), an antioxidant enzyme, contains Zn and Cu in its structure. Superoxide, the substrate of SOD, reacts with fatty acids in the membranes to form products such as malondialdehyde (MDA) and 4 hydroxy nonenal. These are mutagenic and cellular destructive toxins [18]. Serum Zn and Cu values can give information about antioxidant capacity. Low serum Cu and Zn may contribute to the development of BPPV, as it will reduce the antioxidant capacity. There is no previous research in the literature to explain the relationship between Cu/Zn and vertigo. In our study Cu levels were significantly lower in the BPPV group compared to the control group (p <0.05). This low level supports our hypothesis

that the SOD enzyme increases free radical formation by decreasing its activity and that it triggers BPPV as a result of oxidative stress and Ca metabolism. However, the Zn level was lower in the control group, and this result was not suitable for our hipotesis. Our research results, which show a very positive, strong and statistically significant correlation between Cu and Total Thiol and Native Thiol variables, support the thiol and copper differences between the groups (Tables 5 and 6).

If the increase in oxidative stress cannot be suppressed by the antioxidant capacity, then the damage caused by free radicals begins to increase. The resulting free radical damage may trigger the formation of diseases in cells, tissues and organs. Thus, it has been previously s-hown that oxidative damage plays a role in the formation of more than one hundred serious diseases [14–18]. Considering our research results, thiol-disulphide balance, which is a strong bidirectional indicator of antioxidant capacity and oxidative stress, can also be considered as a good clinical laboratory marker for BPPV. The increase in disulphide and decrease in thiol seem to be possible parts of a biochemical mechanism that triggers the formation of BPPV.

The limitations of our study were the small number of patients ane the one time blood sample was taken from the patients. Blood samples were taken from the patients only one time and it was during the attack, We did not check the control blood after the attack. We also could not measure serum calcium and vitamin D levels. Measuring the serum calcium and vitamin D values could have helped us to interpret the pathophysiology of BPPV more clearly. However, we had to limit our work due to financial inadequacies.In addition, due to the covid-19 pandemic that affected the whole world, a high number of patients could not be established in our study due to the limitation of hospital admissions for non-covid-19 reasons.

## Conclusion

As a result, low Cu and Zn levels decrease the antioxidant activity and increase the formation of BPPV. Serum oxidative stress and trace elements play a role in the pathophysiology of BPPV. We believe that serum trace elements and Thiol-disulfide homeostasis may be a possible laboratory indicator in BPPV. We present the cut-off values for Cu and Zn in vertigo patients for the first time in the literature. We think that these cut-off values of the trace elements and thiol/disulfide hemostasis can be used clinically by physicians in the etiology, diagnosis and treatment of vertigo. We suggest researchers to emphasize these results more strongly with the findings of a higher number of patients and samples.

## Supporting information

**S1 File.**
(XLSX)

**S2 File.**
(DOCX)

## Author Contributions

**Conceptualization:** Hüseyin Günizi.

**Data curation:** Hüseyin Günizi, Hasan Basri Savaş.

**Methodology:** Hüseyin Günizi.

**Project administration:** Hüseyin Günizi.

**Software:** Hüseyin Günizi, Hasan Basri Savaş.

**Writing – original draft:** Hüseyin Günizi.

**Writing – review & editing:** Hüseyin Günizi.

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
