## [Decision Letter · Decision Letter 0]

13 Feb 2022

PONE-D-21-36104An Evaluation of Trace Elements and Oxidative Stress in Patients With Benign Paroxysmal Positional VertigoPLOS ONE

Dear Dr. gunizi,

Thank you for submitting your manuscript to PLOS ONE. After careful consideration, we feel that it has merit but does not fully meet PLOS ONE’s publication criteria as it currently stands. Therefore, we invite you to submit a revised version of the manuscript that addresses the points raised during the review process.

We look forward to receiving your revised manuscript.

Kind regards,

Nicolás Pérez-Fernández

Academic Editor

PLOS ONE

Journal Requirements:

Reviewers' comments:

Reviewer's Responses to Questions

**Comments to the Author**

1. Is the manuscript technically sound, and do the data support the conclusions?

Reviewer #1: Yes

Reviewer #2: Yes

2. Has the statistical analysis been performed appropriately and rigorously? 

Reviewer #1: Yes

Reviewer #2: Yes

3. Have the authors made all data underlying the findings in their manuscript fully available?

Reviewer #1: Yes

Reviewer #2: Yes

4. Is the manuscript presented in an intelligible fashion and written in standard English?

Reviewer #1: Yes

Reviewer #2: No

5. Review Comments to the Author

Reviewer #1: In this study, the authors have attempted to determine the role of thiol/disulfide homeostasis as a novel marker of oxidative stress in BPPV.

Although the study is interesting and presents some novel information that deserve to be highlighted, it is necessary to make some changes in the text detailed below:

-Level of evidence must be changed to II-2.

-Introduction: It would be advisable for the authors to further develop the concepts related to oxidative stress and chronic inflammatory diseases already described in the literature. Besides, more bibliography should be provided on the antioxidant effect of the trace elements Cu and Zn.

-Materials and Methods: The selection of patients should include in more detail at what time of the attack the analytical blood extraction was performed and above all, it would be very important to know if in the inclusion criteria, the patients were carried out an analytical study that allows to know if there is a basal inflammatory disease (erythrocyte rate sedimentation, Reactive Protein C, etc ...), in order to rule out from the sample an associated systemic disorders. Furthermore, Erel et al have described association between high levels disulphide values in plasma of smoking, diabetic and obesity patients.

Measurement of Thiol and Disulphide homeostasis, the text must be improved because in Tables: Reduced Thiol, oxidized thiol and thiol oxidation/reduction ratio are showed.

-statistical analysis: In the non-parametric study of the means by means of the Mann U test, I understand that it would be advisable to add the Bonferroni correction and if it is not necessary it should be expressed in the text.

-Results: The text: "The risk of developing BPPV in patients with serum NT level higher than 416.5 μmol/L was 85,615 times higher than those with serum NT level lower than 416.5 μmol/L (Odds ratio: 85.615 %95 CI :23.160-316.499 P<0.001)."

Reviewing Table 4, I understand that it is an error since in the table it is expressed that values lower than 416.5 are those that are associated with an odds ratio 85,615 times higher than those with values higher than that figure.

Dscussion:The authors discuss the possible role of oxidative stress in calcium and vitamin D metabolism but in the Material and Methods of the present series have not led to any studies related to it. This topic must be explained or put into limitations of the study.

Limitations should be extended in the corresponding final section

Reference 20. could be completed with another by Kundi H, Ates I, Kiziltunc E, Cetin M, Cicekcioglu H, Neselioglu S, Erel O, Ornek E. A novel oxidative stress marker in acute myocardial infarction; thiol/disulphide homeostasis. Am J Emerg Med. 2015 Nov;33(11):1567-71. doi: 10.1016/j.ajem.2015.06.016. Epub 2015 Jun 14. PMID: 26143314.

In Table 3, it is expressed that the elements Zn and Cu, as well as the total thiol show values lower than 0.75 in the area under the ROC curve, and this finding must be interpreted as being a regular test or perhaps not completely adequate to detect what the authors are looking for. In addition, it is necessary that the values are expressed in a new Figure in order to graphically assess the results of the ROC curve.

Reviewer #2: Dear authors, congratulations for your research!

There are some corrections and clarifications, that I would like to propose:

1.in the abstract> abbreviation for BPPV not BPVV

2. Introduction> instead of "BPPV occurs as a result of endolymph flow that occurs due to the free movement.." Better, BPPV is ascribed to otoconial matter dislodged from utricular macula and attached to the cupula of the affected semicircular canal (cupulolithiasis) or free-floating within its lumen (canalolithiasis).

3. Material and Methods > Epley's maneuver was performed for the posterior SCC-BPPV, Ok . How about the lateral SCC BPPV ? What is the percentage of SCC affected ? please clarify

4. results> Is there any difference of Cu level between gender in your study group? please clarify

5. Discussion > This article "G. Zucca, S. Valli, P. Valli, P. Perin and E. Mira, Why do benign paroxysmal positional vertigo episodes recover spontaneously? J Vestib Res 8 (1998), 325–329." show us that is pivotal the calcium concentration in that patients. Did you measure ?

TDD > what does it mean?

6. PLOS authors have the option to publish the peer review history of their article (what does this mean?). If published, this will include your full peer review and any attached files.

Reviewer #1: No

Reviewer #2: No

---

## [Author Response · Author response to Decision Letter 0]

13 Apr 2022

Reviewer 1 Comments and Author’s answer:

 -Level of evidence must be changed to II-2.

The specified adjustment has been made.

-Introduction: It would be advisable for the authors to further develop the concepts related to oxidative stress and chronic inflammatory diseases already described in the literature. Besides, more bibliography should be provided on the antioxidant effect of the trace elements Cu and Zn.

Introduciton updated with new references 

‘’Oxidative stress is a condition caused by an imbalance between production and accumulation of oxygen reactive species (ROS) in cells or tissues and the ability of a biological system to detoxify these reactive products.(yeni1) Superoxide radicals, hydrogen peroxide, hydroxyl radicals and singlet oxygen are commonly defined reactive oxygen species (ROS).6-7 In the tissues, high rates of oxygen utilization lead to the generation of partially reduced forms of oxygen or reactive oxygen species (ROS), which are derived from the activity of electron transport chain in mitochondria.8 Free radicals cause oxidative cell damage and this is a situation that occurs continuously during metabolic processes. ROS and free radicals react with and damage several cellular structures such as proteins, lipids, membranes, lipoproteins, and deoxyribonucleic acid.9 ROS react with most biomolecules, their major targets include unsaturated lipids and intracellular thiols.8 In oxidative stress, thiol oxidizes and converts to disulfide. Thiol balance is an important and dual indicator of oxidant-antioxidant balance.10 Prevention of oxidative damage, keeping it in balance and maintaining vitality is only possible with antioxidant activity. If this process gets out of control it can induces several chronic inflammatory diseases as rheumatoid arthritis, and degenerative, body aging process and acute pathologies as stroke.9,11

Trace elements constitute the building blocks of enzymes, hormones and vitamins as well as metabolism support, healthy tissue formation and supporting the immune system. Zn and Cu are the trace elements that are essential and vital for the body. Zinc is an enzymatic activator and catalyst in hormonal processes and amino acid cycle. Copper is a necessary trace element in the controlled work of enzymes in biological oxidation events. The extracellular form of the superoxide dismutase enzyme, which is the only antioxidant secreted by fibroblast, glia and endothelial cells and capable of inactivating superoxide radicals enzymatically at the extracellular level, contains Cu and Zn. It has a very important role in the prevention of many diseases such as oxidant damage, inflammation and fibrosis.12 Therefore, the antioxidant effect of Zn and Cu is remarkable.’’

-Materials and Methods: The selection of patients should include in more detail at what time of the attack the analytical blood extraction was performed and above all, it would be very important to know if in the inclusion criteria, the patients were carried out an analytical study that allows to know if there is a basal inflammatory disease (erythrocyte rate sedimentation, Reactive Protein C, etc ...), in order to rule out from the sample an associated systemic disorders. 

The criteria for inclusion and exclusion from the study were more clearly stated.

‘’Dix-Hallpike and Supine Head-Roll maneuvers were performed. Epley's and Barbecue repositioning maneuver was applied to patients who developed rotatory nystagmus during the maneuvers and supported BPVV. Among these patients, those who did not have diabetes mellitus, hyperlipidemia, etc., had normal parameters such as sedimentation and c-reactive protein in routine blood tests, and had a first BPPV attack were included in the study. Blood samples were taken from patients diagnosed with BPVV to measure serum Zn and Cu levels and oxidative stress levels during at the first attack and before 12:00 a.m. while they were fasting. The control group included 66 healthy volunteers with age and gender distribution similar to the BPPV patient group. They had no neurotologic symptoms and no history of any dizziness or imbalance.’’

Measurement of Thiol and Disulphide homeostasis, the text must be improved because in Tables: Reduced Thiol, oxidized thiol and thiol oxidation/reduction ratio are showed.

The measurement of Thiol and Disulfide homeostasis has been clarified more clearly. 

‘’Thiol / disulfide homeostasis measurements were made using the standard colorimetric method described in the literature.10 Two separate serum samples were collected for each patient. Serum total thiol measurement was performed using standard colorimetric method and commercial kit (Rel Assay Diagnostics, Gaziantep). Ten µL sample was treated with 10µL sodium borohydride in 50% methanol-water solution (v/v; R1). 110µL 6.715mM formaldehyde and 10.0mM ethylenediaminetetraacetic acid (EDTA) in Tris buffer 100mM (pH 8.2) used for Excess reductants eliminating to determine total thiol. A 10µL sample was treated with 10µL sodium chloride in 50% methanol-water solution (v/v; R1’) and 110µL 6.715mM formaldehyde and 10.0mM EDTA in Tris buffer 100mM (pH 8.2) used to determine native thiol. Dynamic disulfide amount was calculated by taking half of the difference between the total thiol and natural thiol groups. All the chemicals were purchased from Merck Chemicals (Darmstadt, Germany) and Sigma-Aldrich Chemie (Milwaukee, Wisconsin, USA). In this way, natural and total thiols were calculated. Then serum disulfide levels and disulfide / natural thiol, disulfide / total thiol and natural / total thiol ratios were determined. Units are given as μmol/L .10’’ 

-statistical analysis: In the non-parametric study of the means by means of the Mann U test, I understand that it would be advisable to add the Bonferroni correction and if it is not necessary it should be expressed in the text.

Since we only looked at 2 parameters, Bonferroni correction was not applied and this material method was also specified.

-Results: The text: "The risk of developing BPPV in patients with serum NT level higher than 416.5 μmol/L was 85,615 times higher than those with serum NT level lower than 416.5 μmol/L (Odds ratio: 85.615 %95 CI :23.160-316.499 P<0.001)."

Reviewing Table 4, I understand that it is an error since in the table it is expressed that values lower than 416.5 are those that are associated with an odds ratio 85,615 times higher than those with values higher than that figure.

The wrong in the sentence was understood and corrected.

‘’The risk of developing BPPV in patients with serum NT level lower than 416.5 μmol/L was 85,615 times higher than those with serum NT level higher than 416.5 μmol/L (Odds ratio: 85.615 %95 CI :23.160-316.499 P<0.001).’’

Dscussion:The authors discuss the possible role of oxidative stress in calcium and vitamin D metabolism but in the Material and Methods of the present series have not led to any studies related to it. This topic must be explained or put into limitations of the study.

Calcium and vitamin D were used in the discussion because of their association with oxidative stress and BPVV. This topic has been explained and discussed in more detail.

Limitations should be extended in the corresponding final section

Limitations extended 

‘’The limitations of our study were the small number of patients ane the one time blood sample was taken from the patients. Blood samples were taken from the patients only one time and it was during the attack, We did not check the control blood after the attack. We also could not measure serum calcium and vitamin D levels. Measuring the serum calcium and vitamin D values could have helped us to interpret the pathophysiology of BPVV more clearly. However, we had to limit our work due to financial inadequacies.In addition, due to the covid-19 pandemic that affected the whole world, a high number of patients could not be established in our study due to the limitation of hospital admissions for non-covid-19 reasons.’’

Reference 20. could be completed with another by Kundi H, Ates I, Kiziltunc E, Cetin M, Cicekcioglu H, Neselioglu S, Erel O, Ornek E. A novel oxidative stress marker in acute myocardial infarction; thiol/disulphide homeostasis. Am J Emerg Med. 2015 Nov;33(11):1567-71. doi: 10.1016/j.ajem.2015.06.016. Epub 2015 Jun 14. PMID: 26143314.

Corrections stated by the referee in the reference section were made.

In Table 3, it is expressed that the elements Zn and Cu, as well as the total thiol show values lower than 0.75 in the area under the ROC curve, and this finding must be interpreted as being a regular test or perhaps not completely adequate to detect what the authors are looking for.

This was highlighted in the results. 

‘’ In the ROC analysis of Cu, Zn and Total Thiol values, the AUC values were below 0.75, so we could not evaluate these values as independent predictive factors.’’

 In addition, it is necessary that the values are expressed in a new Figure in order to graphically assess the results of the ROC curve.

Added a figure showing the ROC curve.

Reviewer 2’s comments and Author’s answers:

1.in the abstract> abbreviation for BPPV not BPVV

Changed the abbreviation BPVV to BPPV in the summary section.

2. Introduction> instead of "BPPV occurs as a result of endolymph flow that occurs due to the free movement.." Better, BPPV is ascribed to otoconial matter dislodged from utricular macula and attached to the cupula of the affected semicircular canal (cupulolithiasis) or free-floating within its lumen (canalolithiasis).

The previous sentence, the specified sentence has been changed.

3. Material and Methods > Epley's maneuver was performed for the posterior SCC-BPPV, Ok . How about the lateral SCC BPPV ? What is the percentage of SCC affected ? please clarify

This sentence has been added to the results ‘’There were 54 (82%) patients with BPPV affecting the posterior and 12(18%) patients with BPPV affecting the lateral canal’’

4. results> Is there any difference of Cu level between gender in your study group? please clarify

This sentence has been added to the results ‘’There were no statistically significant differences between male and female genders in terms of Cu and Zn levels.’’

5. Discussion > This article "G. Zucca, S. Valli, P. Valli, P. Perin and E. Mira, Why do benign paroxysmal positional vertigo episodes recover spontaneously? J Vestib Res 8 (1998), 325–329." show us that is pivotal the calcium concentration in that patients. Did you measure ?

This subject was discussed again by adding the mentioned article and the subject was clarified.

‘’ Zucca et al. In their study with the frog inner ear, they reported that in order to dissolve the otoconia in the endolymph, the calcium level in the endolymph should be low, and on the contrary, if the calcium in the endolymph increases, the otoconi cannot be resolved. They stated that high calcium provides a long duration of BPVV attack.20 This fact may contribute to explain the role of increased calcium in the inner ear due to oxidative stress in the pathophysiology of BPPV. In a study, they found that serum vitamin D levels in BPPV patients were significantly lower than in the control group. They suggested that oxidative stress increased and BPVV developed in patients with vitamin D deficiency due to the relationship of vitamin D to Ca metabolism.21 The fact that vitamin D is directly related to calcium physiology makes our hypothesis strong. However, in our study, we could not measure serum calcium and vitamin D values in our patients due to financial reasons. We think that the increased amount of calcium in the endolymph due to oxidative stress is important in the development of BPVV.’’

TDD > what does it mean?

It was noticed that TDD was written incorrectly instead of TDH (Thiol / disulfide homeostasis ) and it was corrected.

---

## [Decision Letter · Decision Letter 1]

5 Aug 2022

PONE-D-21-36104R1An Evaluation of Trace Elements and Oxidative Stress in Patients With Benign Paroxysmal Positional VertigoPLOS ONE

Dear Dr. gunizi,

Thank you for submitting your manuscript to PLOS ONE. After careful consideration, we feel that it has merit but does not fully meet PLOS ONE’s publication criteria as it currently stands. Therefore, we invite you to submit a revised version of the manuscript that addresses the points raised during the review process.

We look forward to receiving your revised manuscript.

Kind regards,

Emre Avci

Academic Editor

PLOS ONE

Additional Staff Editor comments: We noted your manuscript is similar to this published work https://pubmed.ncbi.nlm.nih.gov/30256204/. Please note that our policy requires that related works must be discussed and justifications should be provided in terms of the differences  and what contributions you study make to the community. Please discuss this related work in more details.

Reviewers' comments:

Reviewer's Responses to Questions

**Comments to the Author**

1. If the authors have adequately addressed your comments raised in a previous round of review and you feel that this manuscript is now acceptable for publication, you may indicate that here to bypass the “Comments to the Author” section, enter your conflict of interest statement in the “Confidential to Editor” section, and submit your "Accept" recommendation.

Reviewer #1: All comments have been addressed

Reviewer #3: All comments have been addressed

2. Is the manuscript technically sound, and do the data support the conclusions?

Reviewer #1: Yes

Reviewer #3: Partly

3. Has the statistical analysis been performed appropriately and rigorously? 

Reviewer #1: Yes

Reviewer #3: Yes

4. Have the authors made all data underlying the findings in their manuscript fully available?

Reviewer #1: Yes

Reviewer #3: Yes

5. Is the manuscript presented in an intelligible fashion and written in standard English?

Reviewer #1: Yes

Reviewer #3: Yes

6. Review Comments to the Author

Reviewer #1: For me, all the requirements that have been requested have been met. The authors must go back to review the writing in case there is any more writing error.

Reviewer #3: An evaluation of trace elements and oxidative stress in patients with benign paroxysmal positional vertigo.

Authors resulted that low Cu and Zn levels decrease the antioxidant activity and increase the formation of BPPV and serum oxidative stress and trace elements play a role in the pathophysiology of BPPV. Outcomes of the present study would provide useful information for readers. However, data presentation is insufficient for acceptance without revision. The manuscript needs to be revised in the followings.

#Patients with a history of Meniere's disease, vestibular neuritis, acoustic neurinoma, sudden hearing loss, hearing loss on audiometric examination, history of head trauma, and having an otologic surgery were not included in the study.

It is said that MD and SD are caused by stress, too. Why did you exclude these diseases?

If you conclude that the onset of BPPV is due to a decrease in Cu and Zn, isn’t it necessary to compare BPPV and other vertiginous diseases?

Overdose of Zn over a long period of time reduces Cu absorption. Therefore, it may be a physiological reaction. How about this physiological reaction?

#Serum Zn and Cu values can give information about antioxidant capacity. Low serum Cu and Zn may contribute to the development of BPPV.

The relationship between changes in serum Zn and Cu levels and oxidative stress due to vitamin D deficiency is unclear. To compare the article, authors should provide actual data on this relationship.

7. PLOS authors have the option to publish the peer review history of their article (what does this mean?). If published, this will include your full peer review and any attached files.

Reviewer #1: **Yes: **Juan Carlos Amor-Dorado, MD, PhD

Reviewer #3: **Yes: **Hiroshi Inui, M.D., Ph.D.

---

## [Author Response · Author response to Decision Letter 1]

10 Aug 2022

Response to Reviewers

*Additional Staff Editor comments: We noted your manuscript is similar to this published work https://pubmed.ncbi.nlm.nih.gov/30256204/. Please note that our policy requires that related works must be discussed and justifications should be provided in terms of the differences and what contributions you study make to the community. Please discuss this related work in more details.

This article was mentioned in the discussion of our previous article. but it was mentioned in more detail on your suggestion. 

‘’ However, there are very few studies on BPPV and TDH in the literature. Şahin E. et al. found significantly lower SH and high SD rates during an attack. Their study is similar to our study but our study is more comprehensive and valuable due to the large number of patients and the fact that we have measured zinc and copper (23). In addition, the results of Şahin et al.'s study are consistent with the results of our study and support our findings..’’

**#Patients with a history of Meniere's disease, vestibular neuritis, acoustic neurinoma, sudden hearing loss, hearing loss on audiometric examination, history of head trauma, and having an otologic surgery were not included in the study. It is said that MD and SD are caused by stress, too. Why did you exclude these diseases?

If you conclude that the onset of BPPV is due to a decrease in Cu and Zn, isn’t it necessary to compare BPPV and other vertiginous diseases?

It is known that endolymphatic hydrops rather than stress has a role in the etiology of Meniere's disease. this is far from the etiology of BPVV. In addition, since the etiologies of MD and SD were different from BPVV, these patients were evaluated within the differential diagnosis.

***If you conclude that the onset of BPPV is due to a decrease in Cu and Zn, isn’t it necessary to compare BPPV and other vertiginous diseases?

The onset of BPPV is associated with the decrease in Cu and Zn. Comparison of BPPV with other vertigo diseases can be made with additional and new studies. It may be suggested in the discussion to make new measurements to see if there is the same situation in diseases with similar etiology.

****Overdose of Zn over a long period of time reduces Cu absorption. Therefore, it may be a physiological reaction. How about this physiological reaction?

An overdose of Zn over a long period of time reduces Cu absorption. Therefore, copper may be low as a result of a physiological reaction. However, the low levels of zinc and copper in our study do not match the physiological picture. In addition, there was no use of high-dose zinc in our study.

*****#Serum Zn and Cu values can give information about antioxidant capacity. Low serum Cu and Zn may contribute to the development of BPPV. The relationship between changes in serum Zn and Cu levels and oxidative stress due to vitamin D deficiency is unclear. To compare the article, authors should provide actual data on this relationship.

In the study conducted by Socha et al., the decrease in zinc and Total antioxidant status in Alzheimer's patients compared to the control group also supports our study results (2) However, in our study, we could not measure serum calcium and vitamin D values in our patients due to financial reasons. And this topic was mentioned in the article.

1. Şahin E, Deveci İ, Dinç ME, Özker BY, Biçer C, Erel Ö. Oxidative Status in Patients with Benign Paroxysmal Positional Vertigo. J Int Adv Otol. 2018 Aug;14(2):299-303. doi: 10.5152/iao.2018.4756. 

2. Socha K, Klimiuk K, Naliwajko SK, Soroczyńska J, Puścion-Jakubik A, Markiewicz-Żukowska R, Kochanowicz J. Dietary Habits, Selenium, Copper, Zinc and Total Antioxidant Status in Serum in Relation to Cognitive Functions of Patients with Alzheimer's Disease. Nutrients. 2021 Jan 20;13(2):287. doi: 10.3390/nu13020287.

---

## [Decision Letter · Decision Letter 2]

8 Nov 2022

An Evaluation of Trace Elements and Oxidative Stress in Patients With Benign Paroxysmal Positional Vertigo

PONE-D-21-36104R2

Dear Dr. Gunizi,

We’re pleased to inform you that your manuscript has been judged scientifically suitable for publication and will be formally accepted for publication once it meets all outstanding technical requirements.

Kind regards,

Donovan Anthony McGrowder, PhD., MA., MSc

Academic Editor

PLOS ONE

Additional Editor Comments (optional):

Dear Dr. Gunizi,

The manuscript entitled “An Evaluation of Trace Elements and Oxidative Stress in Patients With Benign Paroxysmal Positional Vertigo” was revised in accordance with the reviewers’ comments and is provisionally accepted pending final checks for formatting and technical requirements.

Regards,

Dr. Donovan McGrowder (Academic Editor)

---

## [Editor Report · Acceptance letter]

10 Nov 2022

PONE-D-21-36104R2 

An Evaluation of Trace Elements and Oxidative Stress in Patients With Benign Paroxysmal Positional Vertigo 

Dear Dr. gunizi:

I'm pleased to inform you that your manuscript has been deemed suitable for publication in PLOS ONE. Congratulations! Your manuscript is now with our production department. 

Kind regards, 

on behalf of

Dr. Donovan Anthony McGrowder 

Academic Editor

PLOS ONE